# Axisymmetric Contact Problems for Composite Pressure Vessels

## Aleksander Muc

Department of Physics, Cracow University of Technology, 31-155 Kraków, Poland; olekmuc@mech.pk.edu.pl

**Abstract:** The present analysis is conducted for the evaluation of contact pressures of axisymmetric shells made of laminated composites or functionally graded materials. This class of problems is usually called the Signorini–Fichera problem (unilateral constraints) and can be solved as the lower-bound problem. The numerical solution of this problem is proposed both for symmetric and unsymmetric shell configurations. The first-ply-failure of such structures is considered. It is demonstrated that the failure occurs at the end of the contact area corresponding to the appearance of stress concentration of radial concentrated forces.

**Keywords:** axisymmetric pressure vessels; contact problems; laminated composites; functionally graded materials; first-ply-failure

## 1. Introduction

The efficient modeling of 3D contact problems is still a challenge in non-linear implicit structural analysis. The broader discussion and studies of possible contact problems in mechanics is presented in [1]. In this area, a variety of problems can be formulated and solved:

- 3D static and dynamic analysis [2,3].
- 2D static and dynamic problems [4].

However, it should be pointed out that the correct and accurate solution of the above problems requires a different approach to the classical one due to the existence of unilateral boundary conditions. The mathematical formulation of such problems is carried out with the use of the variational inequalities—see Panagiotopoulos [5], Muc [6].

The importance and complexity of the numerical approach is underlined in different papers [7–12], where various numerical methods have been studied characterisng the application of dual methods, nonlinear programming methods, asymptotic methods and the Ritz method.

The current investigations are carried out for different material properties of structures, i.e., isotropic, orthotropic or laminates [13–18]. It should be mentioned that Lazarev and Kovtunenko [19] considered the 2D Signorini problem for composites bodies with a rigid inclusion.

The derivation of frictionless/friction reaction forces and contact areas is necessary in various practical engineering problems such as, e.g., in isotropic or composite (fibre-reinforced plastic, functionally graded materials) pressure vessels (tanks) having many discontinuities (unilateral contact reactions) on the saddle supports of horizontal cylindrical structures—Figure 1.

The fundamental key to understanding the behaviour of such problems lies in deriving and examination for interface/contact forces that occur between the support and the vessel. Wilson and Tooth [20] proved analytically and numerically that supports have a crucial effect on the stress concentration in the vessel. Hoa [21] demonstrated the results of experimental works in this area.

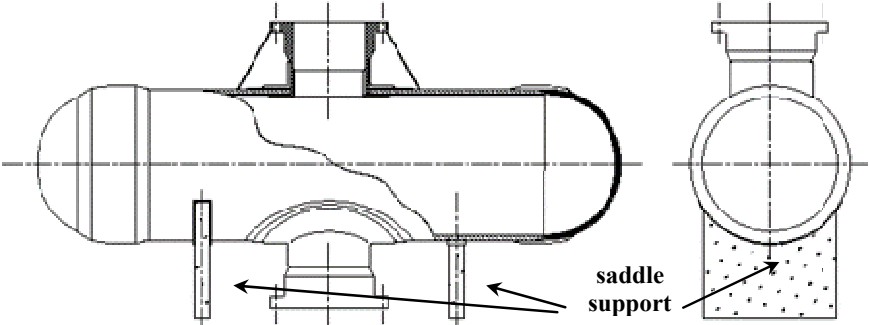

**Figure 1.** Horizontal pressure vessels with two saddle supports.

To explain the problems arising in modeling the contact forces between composite pressure vessels and saddles, the present paper is devoted to the analysis of the interface reaction between axisymmetric composite cylindrical shells and a rigid housing. From the point of view of engineering applications, two fundamental subjects of interest may be distinguished—the contact of the shell: with a rigid or an elastic housing (waterway tunnels [22–26], hydraulic vessels [26]) or with a rigid punch of an arbitrary shape [27,28].

The novelty and achievements of the herein presented problems and numerical results depend on the formulation of the governing relations and demonstration of results in the case of pressure vessels made of FGMs corresponding to an unsymmetric shell configuration.

## 2. Fundamental Relations Describing Deformations of the Axisymmetric Composite (Laminates and FGM) Cylindrical Shells

Let us consider deformations of the axisymmetric cylindrical shell loaded by the internal uniform pressure p—Figure 2. For the first-order transverse shear deformation (FSDT), the general relations are presented by Pielekh, Sukhorolski [24] and Kzhys, Muc [25]. Pipes, tanks, boilers, and various other vessels subjected to internal pressure can be classified as axisymmetrically loaded cylindrical shells. For cylindrical shells of revolution under axisymmetrical loads, the strain displacement relations take the following form:

$$\varepsilon_x = \frac{du}{dx}, \varepsilon_\phi = \frac{w}{R}, \varepsilon_{x\phi} = 0, \varepsilon_{xz} = \beta - \frac{dw}{dx}, \varepsilon_{x\phi} = 0 \tag{1}$$

$$\kappa_x = \frac{d\beta}{dx}, \kappa_\phi = 0, \kappa_{x\phi} = 0, \tag{2}$$

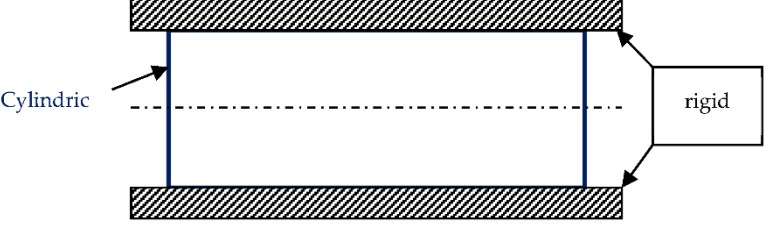

**Figure 2.** Axisymmetric cylindrical shell in the rigid housing.

The symbol $\varepsilon$ denotes the membrane strain and transverse shear strains, $\beta$ is an angle of the rotation of the normal with respect to the shell mid-surface, $\kappa$ means the change in curvature, and 1, 2 corresponds to thelongitudinal $x$ and circumferential directions $\phi$, respectively. $R$ is the radius of the cylinder. The constitutive equations are written in the following way:

$$N_r = A_{rs}\varepsilon_s + B_{rs}\kappa_s, M_r = B_{rs}\varepsilon_s + D_{rs}\kappa_s, r, s = 1, 2 \tag{3}$$

where

$$A_{rs} = \int\limits_{-t/2}^{t/2} Q_{rs}dz, \, B_{rs} = \int\limits_{-t/2}^{t/2} Q_{rs}zdz, \, D_{rs} = \int\limits_{-t/2}^{t/2} Q_{rs}z^2dz, \, r,s = 1,2 \tag{4}$$

Similarly, the transverse shear force vector $\{\widetilde{Q}\}$ is related to transverse shear strains through the constitutive relation as:

$$\widetilde{Q} = xzA_{55}\varepsilon_{xz} \, A_{55} = \frac{5}{4}\sum_{k=1}^{N}(Q_{55})_k\left[t_k - t_{k-1} - \frac{4}{3t^2}\left(t_k^3 - t_k^3\right)\right] \tag{5}$$

For axisymmetric composite structures, the stress–strain relations can be expressed in the following way:

$$\begin{bmatrix} \sigma_x \\ \sigma_\phi \\ \sigma_{xz} \\ \sigma_{x\phi} \end{bmatrix} = \begin{bmatrix} C_{11} & C_{12} & 0 & 0 \\ C_{12} & C_{22} & 0 & 0 \\ 0 & 0 & C_{55} & 0 \\ 0 & 0 & 0 & C_{66} \end{bmatrix} \begin{bmatrix} \varepsilon_x \\ \varepsilon_\phi \\ \varepsilon_{xz} \\ \varepsilon_{x\phi} \end{bmatrix} \tag{6}$$

Using the above relations and the Tsai–Pagano formulation the explicit form of the stiffnesses $A_{rs}$, $A_{55}$, $B_{rs}$, $D_{rs}$ (Eqautions (4) and (5)) can be derived in the classical way both for laminated multilayered structures (Vinson [29]) and porous materials (Kim, Żur, Reddy [30]). The analysis of specially orthotropic specially orthotropic, mid-plane symmetric laminated cylindrical shells subjected to axially symmetric loads all terms {B] = 0 and all other terms ( )$_{16}$ = (.)$_{26}$ = 0—see [29]. For the analysed laminated structures $C_{11} = E_1/(1 - \nu_{x\theta}\nu_{\theta x})$, $C_{12} = \nu_{x\theta}C_{11}$, $C_{22} = E_2/(1 - \nu_{x\theta}\nu_{\theta x})$, $C_{55} = G_{23}$, $C_{66} = G_{12}$, where the symbols $E_1$, $E_2$, $G_{23}$, $G_{12}$, $\nu_{x\theta}$ denote Young's modulus along fibres, Young's modulus in the direction perpendicular to fibres, Kirchhoff's moduli and the Poisson's ratio, respectively. For functionally graded materials

$$C_{11} = C_{22} = E(z)/(1 - \nu^2), Q_{12} = Q_{21} = \nu Q_{11} \tag{7}$$

The elastic modulus $E$ variation characterises the distribution of porosity along the thickness direction $z$ and is defined in the following way:

$$E(z)/E_b = [(E_t/E_b - 1)f(z) + 1] \, f(z) = \left(\frac{z}{t} + \frac{1}{2}\right)^n \tag{8}$$

where the symbols $t$ and $b$ refer to the material properties on top and bottom surfaces, $n$ is power index. $\nu$ is the Poissons ratio.

Writing the equilibrium equations [29]:

$$\frac{dN_x}{dx} = 0 \tag{9}$$

$$\frac{dQ_x}{dx} - \frac{N_x}{R} + p = 0 \tag{10}$$

$$\frac{dM_x}{dx} - Q_x = 0 \tag{11}$$

Assuming that $N_x$ = 0 at the shell edges (boundary conditions) and using the relations (1)–(8) one obtains:

$$A_{55}\left(\frac{d\beta}{dx} - \frac{d^2w}{dx^2}\right) + \frac{R\frac{d\beta}{dx}(A_{12}B_{11} - A_{11}B_{12}) + (A_{12}^2 - A_{11}A_{22})w}{A_{11}R^2} + p = 0 \tag{12}$$

$$-A_{55}\beta + \frac{d^2\beta}{dx^2}\frac{(A_{11}D_{11} - B_{11}^2)}{A_{11}} + \frac{-A_{12}B_{11} + A_{11}B_{12} + A_{11}A_{55}R}{A_{11}R}\frac{dw}{dx} = 0 \tag{13}$$

Eliminating transverse shear effects, i.e., inserting $Q_x = \frac{dM_x}{dx}$ and $\beta = \frac{dw}{dx}$, Equation (12) is reduced to the classical Love–Kirchhoff formulation of axisymmetric cylindrical shell deformations:

$$\frac{R\frac{d^2w}{dx^2}(A_{12}B_{11}-A_{11}B_{12})+(A_{12}^2-A_{11}A_{22})w}{A_{11}R^2} + \frac{B_{12}}{R}\frac{d^2w}{dx^2} - \frac{B_{11}}{A_{11}}\frac{A_{12}}{R}\frac{d^2w}{dx^2}$$
$$+\left[D_{11}-\frac{B_{11}^2}{A_{11}}\right]\frac{d^4w}{dx^4} - p = 0 \tag{14}$$

Galimov [31] and Essenburg, Gulati [32] proved that a shell theory which relaxes the Love–Kirchhoff hypothesis should be used in contact problems. Such an approach allows for the introduction of shearing stress resultants as independent quantities which have essential influence on distribution of contact pressure (see for instance [33]).

### 3. Signorini–Fichera Unilateral Constraint Problem

In our case, the total boundary of the axisymmetric shell is made up of two parts ~$\partial\Omega$ and S (a contact area). On $\partial Q$ the bilateral boundary conditions (classical, i.e., in an equality form) are prescribed. This type of condition will be determined further during the analysis of a particular numerical problem. The second type of boundary condition—the unilateral one, is formulated on the contact domain S (unknown in advance) and it takes the following form:

$$\text{radial displacement } w < 0 \text{ then } S_N = 0 \text{ for } x\notin S \tag{15}$$

$$\text{radial displacement } w \geq 0 \text{ then } S_N + g(w) = 0 \text{ for } x\in S \tag{16}$$

$S_N$ is the normal (radial) reaction of the foundation and $x$ is the longitudinal coordinate. The graphs of functions $g(w)$ considered in this work are plotted in Figure 3. The first diagram (Figure 3a) represents the case of a rigid foundation whereas the second to an elastic foundation of the Winkler type. More possible definitions of the contact normal reactions of the housing $g(w)$ are discussed by Kerr [34].

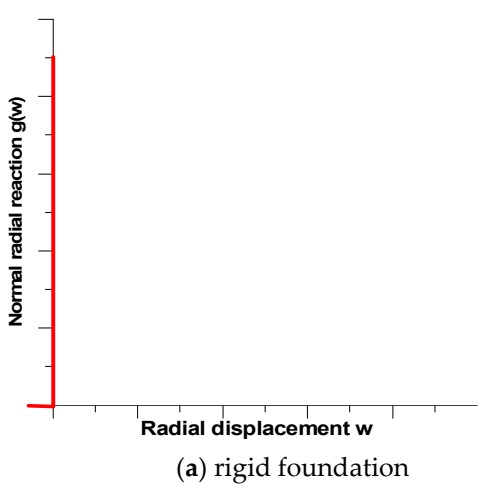

(**a**) rigid foundation

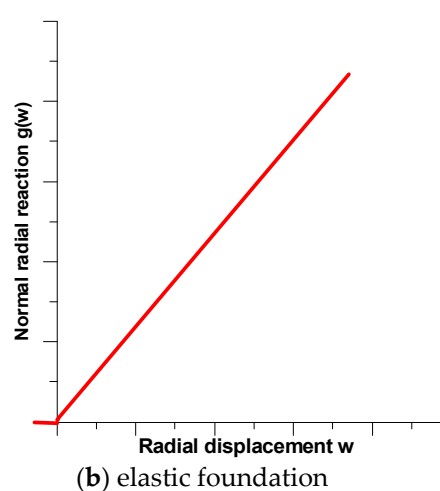

(**b**) elastic foundation

**Figure 3.** Forms of normal reaction $S_N$.

The units in Figure 3 are determined by the formulation of Equations (15) and (16). The function $g(w)$ defines the body forces and has the appropriate units, i.e., Pa, kPa, MPa, etc. The units of the displacement $w$ are classical, i.e., mm, cm, m, etc. The units can be found directly from the relations of the total potential energy functional discussed in Panagiotopoulos [5], Muc [26].

The solution of the contact problem determined consists of finding the stationary points of the total energy on the space of kinematically admissible displacements (i.e., sat-

isfying the strain–displacements relations (1)–(8), the unilateral boundary conditions on S (15) and (16) and some bilateral/boundary conditions). Strictly speaking, the proof of the existence and uniqueness of solutions of the problem described by the functional base on the notions of convexity (for linearly elastic bodies) or poly convexity (for non-linearly elastic bodies). The detailed presentation of those problems is demonstrated in [5,6,26,35].

### 4. Deformations of Axisymmetric Composite Cylindrical Shells Subjected to Simply-Supported Conditions

Let us note that in 2D approach the structural deformations are functions of the geometry (the thickness to radii ratio, the length to radii ratio), composite configurations, material properties and boundary conditions. Assuming the simply supported edges of the axisymmetric shell, i.e.:

$$w = 0, M_x = 0 \text{ at the edges} \tag{17}$$

one can obtain the analytical solutions of Equations (12)–(14)—see [29]. The mentioned relations are derived under the additional boundary condition $N_x = 0$ at the edges.

Considering the deformations, two special cases can be analysed:

- Specially orthotropic mid-plane laminated symmetric structures [29]—symmetric shell geometry, laminate configuration and external loads.
- Functionally graded materials—symmetric shell geometry and external loads but unsymmetric configuration of layered porous materials [36].

The detailed discussion of the influence of symmetric and unsymmetric configuratioms of composites on structural deformations and optimal design is presented by Muc, Flis [37].

In the first case for simply supported shells and using the Love–Kirchhoff hypothesis [29] the radial displacement $w(x)$ have the following form:

$$w(x) = \frac{p(1 - \nu_{x\theta})}{4\varepsilon^4 D_{11}}[1 - \exp(-\varepsilon x)\cos(\varepsilon x)], \varepsilon^4 = \frac{3(1 - \nu_{x\theta}\nu_{\theta x})}{R^2 t^2}\frac{D_{22}}{D_{11}} \tag{18}$$

Figure 4 demonstrates the distribution of the displacement function in Equation (16). The simple parametric studies show the influence of geometric and material properties but they present that the function is convex and increasing with the growth of the x/L ratio. The similar behaviour exists for different laminate configuration and for shells made of FG materials. The explicit derivation of linear differential Equations (10)–(12) can be done easily with the use of Mathematica package—the procedure DSolve.

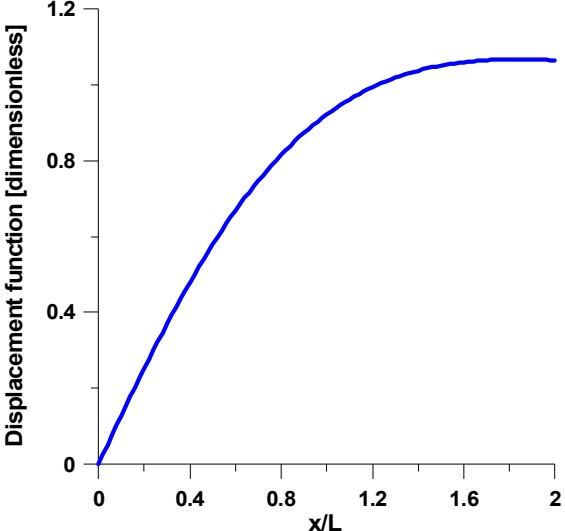

**Figure 4.** Form of the displacement function $1 - \exp(-\varepsilon x)\cos(\varepsilon x)$—isotropic shell t/R = 0.1.

### 5. Modeling of Contact Reactions and Failure Analysis

As it is proved in [5,6,26,33], the unknown in advance normal reactions (16) can be represented in the following way:

$$g(w) = a_1 \exp(a_2 x) + a_3 \exp(a_4 x) + a_5 \delta(x - l_s) \tag{19}$$

where $a_1, \ldots, a_5$ are unknown coefficients and are unknown real constants determined numerically with the aid of the Newton–Raphson procedure. $l_s$ means the point of the separation of the shell from the rigid housing and $\delta(..)$ is the Dirac delta distribution. The necessity of the use of the Dirac functions (circular concentrated forces) is introduced in [26,33]. The problem of the application and description of Dirac's delta distributions is presented by Muc, Zieliński [38].

The verification of the correctness of the approximations of contact reactions and of contact area was made with the use of three methods:

1.  Comparison of the numerical and experimental predictions presented in the literature [20,21,25,26,33];
2.  Evaluation of unilateral contact problems for isotropic structures having the analysed loading and boundary conditions;
3.  Iterative analysis of the influence of the increasing stiffness parameter $\kappa$ in Equation (3) (Figure 3b) in order to observe the possible extension of the contact area and the change of contact reactions.

Now, the classical approach to the contact frictionless or friction problems is based on the finite element analysis. However, for unilateral contact problems, the approximations of contact forces and areas are very sensitive to mesh definitions and, in our opinion, it is better to apply other than FE methods.

Numerical examples are carried out for angle-ply symmetric glass (GFRP) and carbon (CFRP) laminated shells. The material constants of laminates are given in [39] and shown in Table 1. Figure 5 represents the variations of the contact pressures with fibre orientations. For the analysed case, the number of contact area is reduced to one. Comparing these results with the normal radial displacements (Equation (16) and Figure 4), it is seen that the higher stiffness $D_{11}$ results in the increase of the contact pressures.

**Table 1.** Mechanical properties of laminates.

|  | $E_1$ | $E_2$ | $G_{12}$ | $\nu_{12}$ | $X_t$ | $X_c$ | $Y_t$ | $Y_c$ | $S$ |
|---|---|---|---|---|---|---|---|---|---|
|  | In MPa | | | | In MPa | | | | |
| GFRP | 203,000 | 72,000 | 8400 | 0.32 | 3500 | 1540 | 56 | 150 | 60 |
| CFRP | 38,600 | 8270 | 4140 | 0.25 | 1062 | 610 | 31 | 118 | 72 |

Computing the stress distributions along the thickness direction $z$:

$$\sigma_{x(\theta)} = \frac{N_{x(\theta)}}{t} \pm \frac{6M_{x(\theta)}}{t^2} \tag{20}$$

one can find immediately that the maximal stress concentration occurs at the bottom layer of the laminate due the distributions plotted in Figure 5. For the first-ply-failure analysis the Hoffmann criterion is a dominant failure modes for structures made of CFRP, and the Tsai-Wu criterion for structures made of GFRP (see e.g., [40]) due to the stress concentration at the end of contact region—Figure 5.

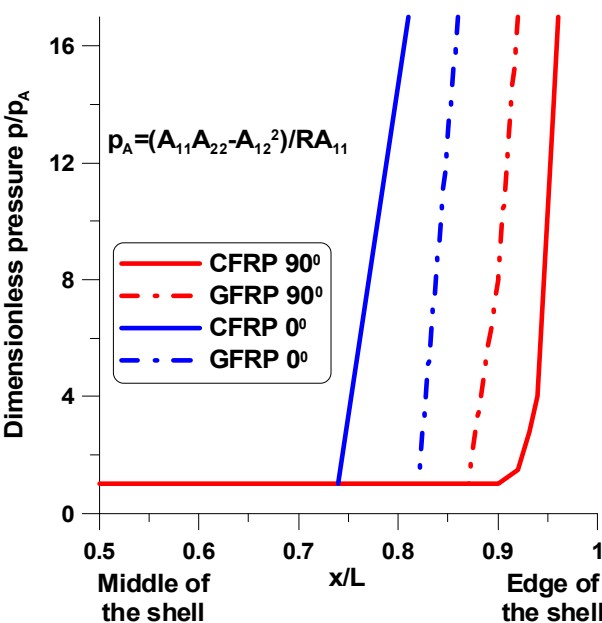

**Figure 5.** Variations of the contact pressures for angle-ply symmetric laminates—L/R = 2.

Damage of structures made of FG materials is studied numerically—see, e.g., [41]. The material gradation results in the difficulty in analyzing failure modes and criteria. The composition of the constituents varies with physical dimension. Figure 6 represents the change of the multiplier $M = \left[ D_{11} - \frac{B_{11}^2}{A_{11}} \right]$ —Equation (12) for different location of constituents and the power index n. It is assumed that the analyzed FG material is composed of the metal Ti (E = 100 [GPa]) and the ceramic TiB (E = 300 [GPa]). The composite material is unsymmetric since $B_{rs}$ terms Equation (4) are not equal to zero.

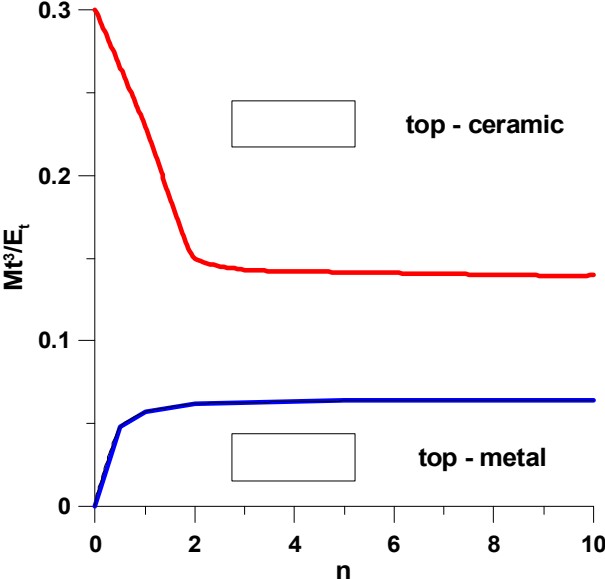

**Figure 6.** Variations of the multiplier $M = \left[ D_{11} - \frac{B_{11}^2}{A_{11}} \right]$ with different location of unsymmetric ceramic/metal components of the structure—L/R = 2.

The distributions of contact reactions for cylindrical vessels made of unsymmetric FGMs are plotted in Figure 7. The form of the reactions depends on the stiffness parameters characterised by the multiplier M. The increase of the M value results in the shift of

the maximal contact concentrated forces to the middle of the shell. The results demonstrate the possibility of the reduction of the stress concentration due to variation of the FGMs constituents.

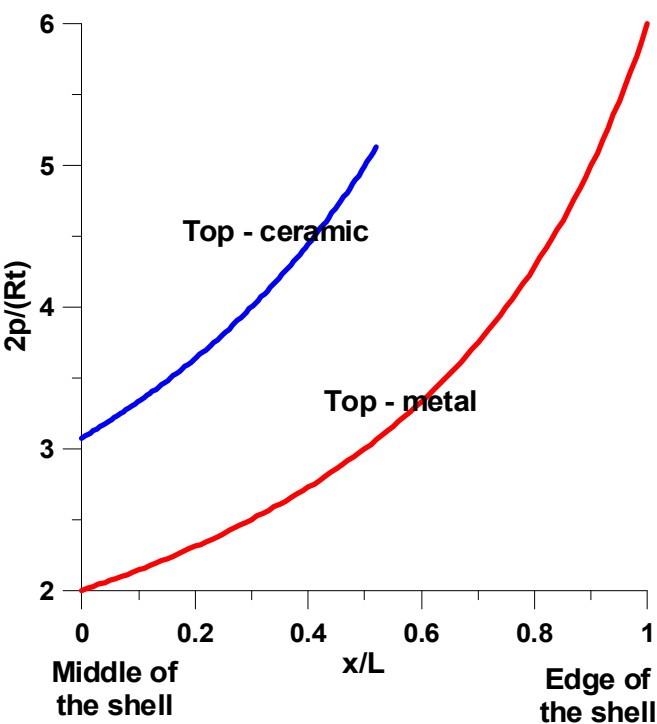

**Figure 7.** Contact reactions with vessels made of unsymmetric FGM layers.

### 6. Conclusions

1. The conducted analysis of axisymmetric contact problem is the extension of my previous analysis in this area. The presented results lead to the following conclusions:
2. The considered shell wall configuration has a significant influence on the form of normal contact reactions as well as on the contact length.
3. For laminated symmetric configuration fibre orientations and mechanical material properties change the form/description of unilateral contact problems.
4. For shells made of functionally graded material, the unsymmetric shell wall configuration variations of the introduced multiplier can represent changes of contact reactions and contact area.

For the proposed semi-analytical methods of computations implemented in the analysis of frictionless unilateral contact problems, the accuracy is higher than for the finite element method, since the results are not dependent on the introduced mesh division.

**Funding:** This research received no external funding.

**Informed Consent Statement:** Not applicable.

**Conflicts of Interest:** The authors declares no conflict of interest.

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
