# Peer review of "Axisymmetric Contact Problems for Composite Pressure Vessels"

_jcs, doi:10.3390/jcs6050143_

Round 1
Reviewer 1 Report
I reviewed the paper and found it is proper paper of com sci however it need major revision as follow.
-the language of the manuscript needs to be polished.
-the manuscript does not present any significant achievement in this type of analysis.
-the figure 1 needs to be more clarify.
-does the author used any mechanical feature with the others litearture hypothesis?
_the validation of ACP performed with other work?
Finally some figures do not present proper clarification for the images and do not disscused well
-
Author Response
The answers are included in the attached file

Reviewer 2 Report
Dear author of the work: “Axisymmetric contact problems for composite pressure vessels”. I find the article interesting and scientifically prepared properly. I have no comments practically on the substantive side of it.
Comments on the editing side of the work which, in my opinion, should be improved:
- Introduction. This part presents Fig. 1 as an indication of the analyzed problem. Please mark the dangerous places covered by the description in this drawing.
- In the next section “2. Deformations of the axisymmetric composite (laminates and FGM) cylindrical shells” The author of the paper present mathematical relationships used to determine deformations. Therefore, I propose to change the title of this chapter. In this chapter, there are mathematical dependencies without clearly indicating whether they are developed on the basis of the literature. I am asking for a correction.
- The next chapter "3. Signorini-Fichera unilateral constraint problem” In my opinion, it should be noted at this point that these are research results. It needs to be changed. The graphs presented in Fig. 3 do not have values ​​on the axes. The range of the variables was not specified. In my opinion this should be improved. Please also mark in this point what is the achievement of the author of the work.
- Chapter 4 Fig. 4 one of the axes has no value.
- Chapter 5 Please standardize the unit in table 1. GPa and MPa were used. I suggest choosing one. Please provide or mark on figures 5-7 with what accuracy the dependencies were determined.
The work should summarize and conclusions that are missing. Please complete the article.
Author Response

(The authors gave the same response as above.)

Round 2
Reviewer 1 Report
The paper is properly revised and can be accepted in this form.
Reviewer 2 Report
Congratulations.In my opinion it is ok